# Identification of Two Subsets of Subcompartment A1 Associated with High Transcriptional Activity and Frequent Loop Extrusion

**DOI:** 10.3390/biology12081058

**Published:** 2023-07-27

**Authors:** Zihang Yin, Shuang Cui, Song Xue, Yufan Xie, Yefan Wang, Chengling Zhao, Zhiyu Zhang, Tao Wu, Guojun Hou, Wuming Wang, Sheila Q. Xie, Yue Wu, Ya Guo

**Affiliations:** 1Sheng Yushou Center of Cell Biology and Immunology, Joint International Research Laboratory of Metabolic and Developmental Sciences, School of Life Sciences and Biotechnology, Shanghai Jiao Tong University, Shanghai 200240, China; yinzihang@sjtu.edu.cn (Z.Y.); cuishuang@sjtu.edu.cn (S.C.); xyf0402@sjtu.edu.cn (Y.X.); yefan.wang@sjtu.edu.cn (Y.W.); zhaochengling@sjtu.edu.cn (C.Z.); zhangzhiyu14@sjtu.edu.cn (Z.Z.); wutao0920@sjtu.edu.cn (T.W.); 2WLA Laboratories, Shanghai 201203, China; 3Department of Bioinformatics and Biostatistics, School of Life Sciences and Biotechnology, Shanghai Jiao Tong University, Shanghai 200240, China; xuesong2019@sjtu.edu.cn; 4Shanghai Institute of Rheumatology, Renji Hospital, Shanghai Jiao Tong University School of Medicine (SJTUSM), Shanghai 200001, China; houguojun225@163.com; 5CUHK-SDU Joint Laboratory on Reproductive Genetics, School of Biomedical Sciences, The Chinese University of Hong Kong, Hong Kong, China; wangwuming@cuhk.edu.hk; 6MRC London Institute of Medical Sciences, London W12 0NN, UK; sheila.xie@lms.mrc.ac.uk; 7Institute of Clinical Sciences, Imperial College London, London W12 0NN, UK

**Keywords:** genome folding, A/B compartments, TADs, contact domains, chromatin looping, tissue-specific gene

## Abstract

**Simple Summary:**

Genomic DNA is folded into chromatin interaction patterns contributing to logical control of gene expression in mammalian cells, but how these highly ordered structures form is not yet fully understood. To assess to what extent functional gene activity is relative to the positioning of genes within the 3D nuclear landscape, we analyzed genome-wide gene expression and chromatin conformation capture data together in five distinct types of cells. We observed that 3D chromatin repositioning frequently occurs during cell differentiation, and these chromatin relocations are significantly associated with changes in gene expression levels. A set of genomic loci with extraordinarily high gene density participates in the establishment of common subcompartment A1 across the genome in all these five cells. By contrast, regulatory genomic segments enriched in cell type-specific genes are engaged in the formation of variable A1. Both subsets of subcompartment A1 bearing the strongest euchromatin signals harbor topological domains with frequent intradomain interactions to facilitate gene regulation. Thus, our study links the gene transcriptional levels with their subcompartment positioning, suggesting a key role of both constitutive and regulatory transcriptional activity in the 3D genome organization.

**Abstract:**

Three-dimensional genome organization has been increasingly recognized as an important determinant of the precise regulation of gene expression in mammalian cells, yet the relationship between gene transcriptional activity and spatial subcompartment positioning is still not fully comprehended. Here, we first utilized genome-wide Hi-C data to infer eight types of subcompartment (labeled A1, A2, A3, A4, B1, B2, B3, and B4) in mouse embryonic stem cells and four primary differentiated cell types, including thymocytes, macrophages, neural progenitor cells, and cortical neurons. Transitions of subcompartments may confer gene expression changes in different cell types. Intriguingly, we identified two subsets of subcompartments defined by higher gene density and characterized by strongly looped contact domains, named common A1 and variable A1, respectively. We revealed that common A1, which includes highly expressed genes and abundant housekeeping genes, shows a ~2-fold higher gene density than the variable A1, where cell type-specific genes are significantly enriched. Thus, our study supports a model in which both types of genomic loci with constitutive and regulatory high transcriptional activity can drive the subcompartment A1 formation. Special chromatin subcompartment arrangement and intradomain interactions may, in turn, contribute to maintaining proper levels of gene expression, especially for regulatory non-housekeeping genes.

## 1. Introduction

In a typical mammalian cell, total genomic DNA exceeds two meters in length, but needs to be folded into a cell nucleus with a diameter of approximately 10 μm in a highly ordered manner [1,2,3]. It has been shown that chromosomes occupy distinct regions in three-dimensional (3D) spaces in the interphase cell nucleus, called chromosome territories [3,4,5], and are further organized into compartments A and B [6], or more sophisticated subcompartments [7,8]. During cell differentiation, chromatin undergoes compaction or decompaction, often referred to as compartment positioning, involved in gene regulation [9,10,11]. Compared to compartment B, which corresponds to condensed heterochromatin and gene-poor regions, compartment A is more associated with actively transcribed euchromatin and gene-rich regions [6]. Six different subcompartments (A1, A2, B1, B2, B3, and B4) were originally identified using ultra-high resolution Hi-C data based on unsupervised machine learning algorithms in human lymphoblastoid cells. Subcompartments A1 and A2, corresponding to genomic regions in compartment A, are enriched in highly expressed genes and associated with activating chromatin marks such as H3K27ac, H3K4me3, and H3K4me1. In contrast, the other four subcompartments (B1, B2, B3, and B4) are correlated with genomic loci in compartment B, containing histone marks for gene silencing [7]. In more recent studies, various computational approaches have been developed to infer subcompartments, and transcriptional activity has been shown to be associated with these assigned subcompartments [8,12,13]. Compared to inactive compartment B, genomic regions of compartment A typically contain more complicated chromatin interaction patterns, including looped chromatin domains mainly containing topologically associating domains (TADs) [10,14] and contact domains [7], as well as specific chromatin conformation, such as chromatin loops [7], stripes [15], and jets [16]. These hierarchically organized patterns are primarily formed by the ring-shaped protein complex of cohesin in interphase cells [17,18,19,20,21,22,23]. Although some transcription factors and transcriptional coactivators play essential roles in the establishment of compartment A [24,25,26,27,28], the dominant formation mechanisms of active chromatin compartmentalization, especially for subcompartment organization, are still largely unknown.

Although cohesin has been shown to play important roles in DNA replication [29], DNA repair [30,31,32], and cohesion between sister chromatids [33,34], the most well-known role is the establishment of higher-order chromatin structure in the interphase nuclei, known as cohesin traffic pattern modulated by CTCF and RNAPII [35]. Extruding cohesin can be blocked at a pair of CTCF (CCCTC-binding factor) sites with convergent motif orientation, called “CTCF code” or “CTCF convergence rule” [7,36,37,38,39,40,41]. This rule can be used to define specific chromatin loops or to encode topological domains, which ensure proper enhancer-promoter interactions and prevent improper cross-domain contacts to guarantee precise gene regulation [36]. Indeed, cohesin has a crucial role in the regulation of tissue-specific and inducible functional genes whose activation is exquisitely dependent on cohesin-based interactions [42,43,44]. Apart from CTCF, transcription start sites of housekeeping genes, but not tissue-specific ones, are also frequently located at domain boundaries [10]. Consistently, start sites of transcriptionally active genes can form cohesin extrusion boundaries, though these barriers are not stationary [35,45].

It is thus proposed that both subcompartment arrangement and formation of specific cohesin traffic patterns, which can be encoded by linear genome sequences, contribute to precise control of gene expression. However, spatial distribution characteristics of different types of genes containing housekeeping and cell type-specific genes are still not fully investigated in multiple cell types on a genome-wide scale. To assess to what extent gene transcriptional activity can be associated with their spatial positioning, we initially divided the genome into eight distinct subcompartment types in five different cell types. We observed that gene expression levels were tightly associated with their subcompartment arrangement. Based on the consistency of subcompartment types annotated in these five cell types, we identified a set of gene-dense genomic segments containing abundant housekeeping and other active genes, and another different set of genomic regions enriched with cell type-specific genes. In addition, both subsets of subcompartments exhibit strongly looped domains that facilitate proper gene regulation. Thus, our study reveals the relation between gene regulation and 3D subcompartment positioning during cell differentiation, further indicating a fundamental role of transcriptional activity in active subcompartment formation and topological domain organization.

## 2. Materials and Methods

### 2.1. Experiments

#### 2.1.1. Cell Culture

K562 cells were cultured in Iscove’s Modified Dulbecco’s Medium (Gibco, Waltham, MA, USA) supplemented with 10% (*v*/*v*) fetal bovine serum (Gibco) and 1% penicillin-streptomycin. The HeLa cells were cultured in Dulbecco’s Modified Eagle Medium (Gibco) supplemented with 10% (*v*/*v*) fetal bovine serum (Gibco) and 1% penicillin-streptomycin. Cells were maintained in an incubator with 5% (*v*/*v*) CO_2_ at 37 °C. The K562 (accession number: SCSP-5054) and HeLa (accession number: TCHu187) cell lines were provided by Cell Bank, Chinese Academy of Sciences.

#### 2.1.2. Cleavage under Targets and Tagmentation (CUT&Tag)

CUT&Tag libraries were prepared using the Hyperactive in situ ChIP Library Prep Kit for Illumina (TD901, Vazyme Biotech, Nanjing, China). Briefly, cells were counted, and 100,000 cells were used for each assay. These cells were bound to the conA beads and incubated with anti-H3K27ac (ab4729, Abcam, Cambridge, UK) or anti-Nipbl antibody (A301-779A, Bethyl Laboratories, Montgomery, TX, USA) for 2 h at room temperature. After washing, cells were incubated with a secondary antibody (Goat Anti-Rabbit IgG, Sangon, Shanghai, China), and diluted (1:100) in the DIG Wash buffer for 1 h on a shaker with slow rotation at room temperature. The Hyperactive pG-Tn5 Transposase was then added, and cells were incubated for 1 h. After washing three times with the Dig-300 buffer, these cells were resuspended in the tagmentation buffer and incubated for 1 h at 37 °C. Chromatin was purified using a standard phenol/chloroform/alcohol extraction method. Fragmented DNA was then amplified by PCR and then purified by magnetic beads. The final libraries were sequenced on an Illumina platform.

#### 2.1.3. Chromatin Immunoprecipitation Sequencing (ChIP-Seq)

ChIP was performed as previously described [16]. Briefly, cells were crosslinked with 1% formaldehyde (*v*/*v*, Thermo Fisher Scientific, Waltham, MA, USA) for 10 min at room temperature, followed by quenching with 125 mM glycine for 5 min. Crosslinked chromatin was then fragmented using a sonicator (Scientz) for 28 cycles (30 s on, 30 s off). The lysate was incubated with an H3K27ac antibody (ab4729, Abcam). Dynabeads Protein G beads (Invitrogen, Waltham, MA, USA) were added the next day for 4 h, and ChIP samples were washed with radioimmunoprecipitation assay (RIPA) buffer. After DNA purification, the ChIP-Seq library was prepared using the NEBNext Ultra™ II DNA Library Prep Kit for Illumina (E7645, NEB, Ipswich, MA, USA) following the manufacturer’s protocol and sequenced on an Illumina platform.

### 2.2. Data Analysis

#### 2.2.1. Hi-C Data Processing

Hi-C data were analyzed using the HiC-Pro pipeline (v.2.11.1) [46]. In brief, raw read pairs were mapped to the mouse reference genome (mm9) using Bowtie2 (v2.3.5.1) [47]. After mapping, Hi-C interaction pairs were converted to input format for Juicebox, and valid chromatin interactions were normalized, and final *.hic* files were created using the Juicer tools (v0.7.5) [48,49]. Topologically associating domains (TADs) were identified using the directionality index (DI) [10].

#### 2.2.2. Assignments of Hi-C Subcompartments

The intrachromosomal contact matrices were retrieved from the .*hic* files using the Juicebox dump with the setting ‘observed KR BP 50000’ [48]. Subcompartments were then inferred by the Calder software package with the parameters of ‘50000 sub_domains = TRUE’ [8]. The number of subcompartments was predetermined, and gene density was used to distinguish between A (higher gene density) and B compartment (lower gene density). Inferred eight subcompartments, including A.1.1, A.1.2, A.2.1, A2.2, B.1.1, B.1.2, B.2.1, and B.2.2, were renamed as A1, A2, A3, A4, B1, B2, B3, and B4, respectively. To calculate the frequency of subcompartment repositioning during cell differentiation, the genome was firstly binned into 50 kb intervals, and the proportion of repositioned subcompartments during differentiation was defined by the proportion of repositioning intervals in a given subcompartment type. Chromosomes X and Y were excluded from downstream analysis in this study.

#### 2.2.3. Calculation of Interchromosomal A/B Ratio

The interchromosomal contacts were generated using the Juicebox dump with the setting ‘observed KR BP 50000’ [48]. For each 50 kb bin, *i*, we calculated score *A_i_*, representing its inter-chromosomal contacts with genomic regions annotated as compartment A, and score *B_i_*, representing that of compartment B. For each subcompartment, SC, $\sum_{i}^{N_{SC}}A_{i}$ was the sum of *A_i_* of all the loci in it (*N_SC_*), and $\sum_{i}^{N_{SC}}B_{i}$ was the sum of *B_i_* of all the loci in the subcompartment. Subsequently,$\sum_{i}^{N_{SC}}A_{i}$ and $\sum_{i}^{N_{SC}}B_{i}$ were normalized by dividing *N_A_* and *N_B_* (the total length of chromatin compartment A or B), respectively. The new factor after normalization was named *C_A_* or *C_B_*. The A/B ratio was, therefore, referred to as *C_A_*/*C_B_*. The equation is as follows:(1)$$\frac{A}{B}Ratio=\frac{\frac{\sum_{i}^{N_{SC}}A_{i}}{N_{A}}}{\frac{\sum_{i}^{N_{SC}}B_{i}}{N_{B}}}=\frac{C_{A}}{C_{B}}$$

#### 2.2.4. RNA-Seq Data Analysis

Raw sequencing reads were downloaded from the NCBI Gene Expression Omnibus (GEO) database repository and aligned to the human reference genome (GRCh37/hg19) with TopHat v2.1.0 (setting of “-N 2 -g 20 -x 60”) [50]. The aligned reads mapped to gene exons were counted by the HTSeq-count (v1.99.2) [51]. Active genes were defined as genes with an FPKM (Fragments per kilobase per million) score higher than 1, and highly expressed genes were genes with an FPKM > 10. Genes with an FPKM > 1 and at least five-fold higher expression levels in a particular cell type compared to the other four types were defined as cell type-specific genes.

#### 2.2.5. Calculation of Domain Strength

The arrowhead module of the Juicer software suite (v0.7.5) [49] was used to identify contact domains using Hi-C contact matrices at the resolution of 5 kb and 10 kb. KR (Knight–Ruiz) balancing method was used for contact matrix normalization. Identified domains at the two resolution levels were merged, and domains with smaller sizes were kept for downstream analysis.

To calculate the strength *S* of the chromatin domain from loci *X* to *Y*, we proceeded through the following calculation steps:(2)$$B_{l}=\frac{\sum_{b=1}^{n}\sum_{i=X_{b}}^{Y_{b}-l}I_{i,i+l}}{\sum_{b=1}^{n}{(Y}_{b}-X_{b}-l)}$$
(3)$$N_{X,Y}=\overset{Y}{\underset{i=X}{\sum }}\overset{Y}{\underset{j=i}{\sum }}{(I}_{i,j}-B_{j-i})$$
(4)$$S_{X,Y}=\frac{N_{X,Y}}{T}$$

For a given matrix resolution, we first computed a one-dimensional normalization array *B* (average contact counts of compartment B between different locus pairs); *n* was the number of B compartments at the given resolution. We iterated through the possible distances *l* between locus pairs in compartment B, and counted the values of *I_i,i_*_+*l*_, defined as the KR (Knight–Ruiz) balanced contact value between loci *i* and *i* + *l*. We calculated *B_l_* by dividing the sum of *I_i,i_*_+*l*_ (both *i* and *i* + *l* are identified in the same B compartment) by the sum of locus pairs (*i*, *i* + *l*) at distance *l* in B compartments, as in Equation (2).

Next, to perform contact strength normalization calculation, we subtract KR balanced contact value between a locus pair (*I_i_*_,*j*_) by the average contact strength of two loci at the same distance in the B compartment (*B_j_*_−*i*_). We went through each locus pair (*i*,*j*) in the chromatin domain and summed their normalized contact strength to obtain *N_X_*_,_*_Y_*, as in Equation (3). Once we calculated these values, the strength of the domain was calculated by averaging *N_X_*_,_*_Y_* over the total number of locus pairs *T* within the domain (Equation (4)).

#### 2.2.6. Three-Dimensional Simulation

Data in hic format were converted to the cooler format by using the hic2cool (https://github.com/4dn-dcic/hic2cool, accessed on 12 January 2023) at the resolution of 25,000 base pairs. The HIPPS-DIMEs software package [52] was then used to simulate three-dimensional chromosome organization. For Figure 1A, the iteration number was set to 10, while for others, it was set to 1000. After converting to the binary g3d format using the g3dtools, three-dimensional chromatin contacts were visualized using the 3D viewer in the WashU Epigenome Browser [53]. The genomic coordinates with distinct subcompartment annotations were indicated with different colors.

#### 2.2.7. ChIP-Seq Data Analysis

ChIP-seq data sources were listed in Appendix A. Raw sequencing reads were mapped to the reference genome using Bowtie (v1.0.0) [54]. Mapped reads were extracted by the SAMtools software package (v1.9) [55] with the setting of ‘-h -F 4’. Duplicated reads were removed using the Picard MarkDuplicates function (v2.26.6) (https://github.com/broadinstitute/picard, accessed on 12 December 2021). The pybigwig (https://github.com/deeptools/pyBigWig, accessed on 5 October 2022) was used to analyze the enrichment of signals.

#### 2.2.8. CUT&Tag Data Analysis

The NGmerge program (v0.3) [56] was used to remove adaptors with options -a -u 41. Raw sequencing data were aligned to the GRCh38 reference genome using bowtie2 (v2.3.5.1) [47]. SAMtools (v1.9) [55] were then used to filter mapped reads, and duplicate reads were removed by the Picard (https://github.com/broadinstitute/picard, accessed on 12 December 2021). Significant peaks were identified using the macs2 v2.1.0 [57] with the cut-off q-value <= 0.05 and the parameter–SPMR.

#### 2.2.9. Data Availability

ChIP-seq and CUT&Tag datasets for Nipbl or H3K27ac generated in this study were deposited at the NCBI Gene Expression Omnibus (GEO) under accession number GSE227868. The housekeeping gene list used in this study was derived from the literature [58]. All high-throughput sequencing datasets used in this paper are listed in Appendix A.

## 3. Results

The 3D genome has been well demonstrated to be organized through two independent folding mechanisms [23]: (a) the cohesin-independent mode in which the genome is folded into fine-scale compartments or more sophisticated subcompartments, closely correlated with chromatin state; and (b) the cohesin-driven loop extrusion leads to the formation of TADs or contact domains, which facilitate enhancer-promoter communications within domains. It is suggested that both subcompartment positioning and formation of specific cohesin-dependent chromatin interaction patterns may contribute to precise control of gene expression.

### 3.1. Subcompartment Relocations Frequently Occur and These Transitions Preferentially Correlate with Changes in Gene Expression Levels

#### 3.1.1. Identification of Subcompartments in Five Distinct Cell Types

To assess the impact of spatial positioning of chromatin on gene transcriptional activity, we first inferred eight subcompartments (labeled A1, A2, A3, A4, B1, B2, B3, and B4) using the Calder algorithm (see Section 2) in five different cell types, including three noncycling cell types (thymocytes, macrophages, and cortical neurons), one sorted G_0_/G_1_ cell type (ES cells), and one primary cell type (NP cells) to minimize the influence of cell cycle stages (Appendix A). We then constructed a 3D model based on chromatin contact maps using the HIPPS-DIMES approach (see Section 2) to analyze the 3D distribution characteristics of these eight distinct subcompartment types. Consistent with the fractal globule polymer conformation [6], we observed that genomic regions of chromosome 2 annotated in the same compartment state (active or inactive) exhibited closer proximity in the 3D space (Figure 1A,B). Notably, separated regions from subcompartment A1, depicted in golden yellow, displayed a tendency to cluster together. Because the intrachromosomal (Figure 1C and Appendix A) and interchromosomal contact maps (Figure 1D and Appendix A) displayed patterns that aligned with the characteristic ‘plaid’ pattern of chromatin subcompartments, we subsequently compared these patterns to subcompartment annotation, respectively. Specifically, stronger chromatin contacts within the A1 subcompartment were observed in both intrachromosomal and interchromosomal maps, but the interactions across different subcompartments were limited (Figure 1C,D). We further compared the A/B ratio (interchromosomal contact signals of compartments A divided by signals of compartments B) in eight subcompartment types, which were identified based on intrachromosomal interaction patterns analyzed using the Calder. Indeed, genomic regions in subcompartments A1 displayed a higher A/B ratio, followed by a gradual decrease from A1 to B4 (Figure 1E,F and Appendix A). Thus, these analyses indicated that the identified eight subcompartment types here are closely related to the Hi-C interaction patterns.

#### 3.1.2. Frequent Occurrences of Subcompartment Relocations during Cell Differentiation

We first plotted the transitions of different subcompartments between embryonic stem cells and the four other differentiated cells and found that subcompartment relocations occurred frequently. Upon transition from embryonic stem cells to the thymocytes, 59.2% of subcompartments underwent repositioning. Specifically, within subcompartment A1, 21.9% of A1 was transferred to A2, 6.8% to A3, 2.5% to A4, 1.6% to B1, 0.7% to B2, 0.6% to B3, and 0.4% to B4, respectively (Figure 2A). Similar repositioning patterns were observed in transitions from embryonic stem cells to macrophages, neural progenitors, or cortical neurons (Figure 2B). The proportion of repositioned subcompartments during differentiation from ES cells to macrophages was 60.7%, as well as 61.4% to NP cells and 65.6% to cortical neurons. Similarly, the repositioning proportion was 51.1% from NP cells to cortical neurons.

Our analysis uncovered interesting observations regarding the variability of subcompartments during transitions. We noted that subcompartments A2-B3 were more variable than subcompartments A1 and B4 in these transitions. We proceeded to calculate the proportion of subcompartment relocations derived from ES cells and showed that 32.53% of the subcompartments A1 and 28.95% of the subcompartments B4 remained in all five different cell types. By contrast, less than 1.3% of subcompartments for the other six subcompartment types (A2-B3) exhibited consistency across the five cell types (Figure 2C). We observed similar repositioning proportions in the other four cell types (Appendix A).

We noted that the length of DNA associated with subcompartments A1 or B4 is greater than that of other types of subcompartments, and the step size of subcompartment repositioning is different for each subcompartment type, suggesting that a bias may be present in the results of persistent proportions. To assess the influence, we merged eight subcompartment types into four types, including A1, A2/A3/A4, B1/B2/B3, and B4. The proportions of unchanged relocations for the four merged subcompartment types are 32.53%, 13.05%, 6.60%, and 28.95%, respectively. This further indicates that part of subcompartment A1 or B4 is preferentially persistent than other subcompartment types.

#### 3.1.3. Expression Levels of Genes Are Associated with Their Subcompartment Positioning

To assess the correlation between subcompartment repositioning and gene expression levels, we performed a comparative analysis of relative expression levels (log 2 transformed FPKM) for total protein-coding genes in different subcompartments. The gene expression levels were highest in subcompartments A1, and gradually decreased from A1 to B4 in all five cell types (Figure 3A and Appendix A; Kruskal-Wallis Test, *p* < 2.2 × 10^−16^), suggesting a strong association between subcompartment positioning and gene transcription activity.

Subsequently, we compared the expression changes of genes relocated during cell differentiation from embryonic stem cells to thymocytes (Figure 3B and Appendix A). Consistent with the repositioning degree, we observed the most significant changes in gene expression during transitions from A1 to B-type subcompartments (B1-B4), as well as any one of the other three subcompartments A (A2-A4) to subcompartments B (B1-B4). We further analyzed the changes in gene expression in response to relocations between any two subcompartments in three cell types, including embryonic stem cells, thymocytes, and cortical neurons. Gene expression levels were predominantly decreased from subcompartments A1 to any of the other three A types, and vice versa (Figure 3C). These findings highlighted the dynamic nature of gene expression during subcompartment repositioning.

### 3.2. Differential Spatial Distribution Characteristics between Housekeeping and Cell Type-Specific Genes

#### 3.2.1. Identification of Common A1 Regions That Are Highly Enriched in Active Genes and Highly Expressed Genes

Given the association between gene expression levels and their subcompartment positioning, we proceeded to analyze the gene density of active and highly expressed genes in these eight subcompartment types. We noted that although the density of active or highly expressed genes was not always consistent across compartments A, which corresponds to actively transcribed euchromatin, on a representative chromosome (chr2), both active and highly expressed genes were enriched in subcompartments A1, as opposed to other subcompartment types (Figure 4A). Indeed, the densities of both active and highly expressed genes were highest in subcompartments A1 in all these cells (Figure 4B).

Due to a significant proportion of subcompartments A1 (32.53%) being observed (Figure 2B), we identified common A1, which was annotated as subcompartment A1 in all five cell types. Surprisingly, the average gene numbers exceeded 17.3 active genes per million base pairs, and more than 4.2 for highly expressed genes. In addition, gene density in these common A1 regions was much higher than in variable A1 regions (1.7–2.9 folds for active genes; 1.5–3.1 folds for highly expressed genes), which were annotated as A1 but not included in the common A1 list (Figure 4B). To further confirm the results, we calculated all gap lengths between active genes in common or variable A1. The violin plots, containing interior box plots, showed smaller gap lengths in common A1 compared to variable A1 in all the five cell types (Appendix A), demonstrating a higher gene density in common A1 than variable A1.

#### 3.2.2. Different Spatial Positioning of Housekeeping and Cell Type-Specific Genes in Distinct Cell Types

Next, we conducted an analysis of the gene density of housekeeping genes in different subcompartments, considering the observation that housekeeping genes frequently overlap with common-A1 regions (Figure 4A). Similar to the patterns observed for active and highly expressed genes, housekeeping genes were enriched in subcompartments A1 significantly (Fisher’s Exact Test, *p* < 10^−16^), but there was no remarkable difference between common A1 and variable A1 (Figure 5). By contrast, cell type-specific genes were significantly located in variable A1 rather than common A1 (Figure 4A and Figure 5B). The odds ratios were between 1.82 and 5.36 (Figure 5B).

These analyses indicated that common A1 with high gene density is enriched with active, highly expressed, and housekeeping genes. Compared to common A1, variable A1 is significantly enriched in cell type-specific genes.

### 3.3. Subcompartments A1 Containing More Housekeeping and Cell Type-Specific Genes Harbor Most Strongly Looped Domains

#### 3.3.1. Both Housekeeping and Cell Type-Specific Genes Exhibit a Tendency to Reside within Contact Domains

Previous studies have shown that housekeeping genes, rather than cell type-specific genes (also called tissue-specific genes), are frequently located at TADs boundaries. We confirmed the results in mouse thymocytes (Figure 6A,B) and then investigated the distribution of the two classes of genes around the boundaries of contact domains, which are defined based on high-resolution Hi-C data and present more precise local interactions. Similar to TADs boundaries, contact domain boundaries were also enriched in housekeeping genes. Notably, the normalized gene counts inside contact domains were higher than nearby outside domain regions (Figure 6C,D). Furthermore, we performed an analysis of gene distribution in common A1 and variable A1. We found a higher presence of both housekeeping and cell type-specific genes within domains. However, it is notable that, in contrast to housekeeping genes, cell type-specific genes were more frequently located in variable A1 rather than common A1 (Figure 6E–G).

#### 3.3.2. Strongly Looped Domains Are Most Frequently Located in Subcompartments A1

To further investigate the differential spatial distribution characteristics of housekeeping and cell type-specific genes, which are shown to rely on both enhancer activity and the frequency of cohesin-dependent enhancer-promoter interactions, we compared their locations with positions of strong enhancer peaks and the arrangement of contact domains. As expected, cell type-specific genes frequently colocalized with super-enhancers, which were responsible for driving the expression of genes involved in defining cell identity, while housekeeping genes were enriched in common A1 (Figure 7A). The majority of both types of genes were inside chromatin domains and were frequently located in strongly looped domains in the region of chromosome 2 in thymocytes and embryonic stem cells (Figure 7A and Appendix A).

We then performed a 3D simulation to analyze the relationship between gene subcompartment positioning and their domain strength and found that strongly looped domains were indeed associated with subcompartments A1, highlighted by golden yellow (Figure 7B). As shown in the zoomed region, nine strongly looped domains containing seven in common A1 and two in variable A1, indicated in Figure 7A, were organized into a domain cluster in the 3D space (Figure 7B). Our analysis showed that subcompartment A1 regions, which were characterized by more strongly looped domains, exhibited higher levels of H3K27ac and Nipbl signals, but not heterochromatin-related H3K9me3 mark (Figure 7C–F).

#### 3.3.3. Chromatin Interaction Strength within Contact Domains Is Associated with Their Subcompartment Positioning

We then further analyzed the enrichment of H3K27ac, Nipbl, and H3K9me3, and contact domain strength in eight subcompartments on a genome-wide scale. Consistent with a previous study in human lymphoblastoid GM12878 cells, signals of active histone mark H3K27ac represented a decrease from subcompartments A1 to B4 in thymocytes; by contrast, H3K9me3 enrichment increased gradually in these subcompartment types (Figure 8A,C). As a crucial cohesin-loading regulator, Nipbl plays an essential role in chromatin domain organization, and its occupancy on chromatin is closely associated with active transcription [23]. We showed that similar to the enrichment of H3K27ac signals, subcompartments A1 had the highest Nipbl occupancy, while B4 contained the lowest Nipbl signals across the genome, suggesting a correlation between subcompartment positioning and cohesin-dependent domain organization (Figure 7C, Figure 8A, and Appendix A). Indeed, the most strongly looped domains were predominantly located in the subcompartments A1, with a gradual decrease from A1 to B4 (Figure 8B,D). We then analyzed the differences in domain strength between common A1 and variable A1. We found that, consistent with the H3K27ac and Nipbl signals, domain strength was also similar between the two subtypes of subcompartments A1, although the average strength values were slightly higher in common-A1 regions (Figure 8B,D). Highly similar results were observed in ES cells and HeLa cells (Appendix A). Thus, these analyses suggest two independent mechanisms contributing to subcompartment A1 formation: (a) persistent transcriptional activity is associated with the steady proximity of active chromatin in 3D space to facilitate the building up of common-A1 structure, and (b) regulatory transcriptional activity is involved in variable-A1 establishment in a cell type-specific manner. Two types of A1 boost intradomain chromatin interactions that are dependent on cohesin-mediated loop extrusion.

Next, we confirmed the results in human cells by analyzing Hi-C and gene expression data. Firstly, active genes were maximally enriched in subcompartments A1 and common-A1 regions indeed harbor more active genes in both human K562 and HeLa cells (Appendix A, Appendix A). Additionally, a similar pattern of Nipbl enrichment in eight distinct subcompartments, measured by CUT&Tag assay, was observed in the two human cells (Figure 8C and Appendix A). Furthermore, subcompartments A1 included more strongly looped domains with a gradually decreasing domain strength from A1 to B4 (Figure 8D and Appendix A). Lastly, highly similar to the thymocytes and embryonic stem cells, human K562 cells also presented highly frequent interactions within contact domains in both common A1 and variable-A1 regions, as well as enrichments of H3K27ac and Nipbl (Figure 8D). Additionally, we observed a very similar pattern of strength using contact domains ranging from 160 to 220 kb, indicating that the trend was independent of domain size.

## 4. Discussion

Nuclear subcompartments, initially defined by ultra-high resolution Hi-C maps, are closely associated with transcriptional activity, as well as the patterns of histone modification. However, the identification of subcompartments by machine learning algorithms using interchromosomal contacts relies on ultra-high-resolution chromatin conformation capture data. In this study, we employed a published algorithm [8], which utilizes intrachromosomal interaction maps with varying total read numbers to infer hierarchies of subcompartments. We analyzed five distinct cell types, and our data showed that the patterns of intrachromosomal interactions exhibited a characteristic “plaid” pattern corresponding to annotated subcompartment types. Moreover, these subcompartments based on local intrachromosomal contacts could largely reflect interchromosomal interaction patterns. Additionally, we observed the dynamic changes of chromatin state from subcompartment A1 to B4, including a decrease of H3K27ac and an increase of H3K9me3 signals. Thus, the main features of subcompartment types annotated in this study are consistent with previous studies [7,12,13].

We further showed that subcompartment repositioning frequently occurred during cell differentiation. However, we noted that some subcompartments specifically annotated as A1 (32.53%) or B4 (28.95%) were extraordinarily persistent in all five distinct cell types, suggesting two differential mechanisms for chromatin compartmentalization upon differentiation. The subcompartment B4 regions have the lowest active gene density and the highest enrichment of H3K9me3 (Figure 8A and Figure 9A), a typical mark for constitutive heterochromatin, suggesting that common B4 consists of a set of genomic regions with condensed chromatin conformation that is believed to be formed by liquid–liquid phase separation [59,60,61,62]. In contrast to B4, the subcompartment A1 was mostly enriched in highly expressed genes and active histone mark H3K27ac (Figure 8A and Figure 9A), associated with a loosely packed euchromatin. We further separated subcompartment A1 into two subsets, common A1 and variable A1. Unexpectedly, common A1 were high-gene-density regions (even approximately two-fold higher than variable A1). Housekeeping genes were distributed in both common and variable subsets of A1, but cell type-specific genes were majorly enriched in variable A1 regions (Figure 9A). Thus, these findings indicate that subcompartment A1 consists of two classes of chromatin: (a) highly transcriptional genomic regions in distinct cell types lead the establishment of constitutive A1; (b) regulatory genomic loci with high transcriptional activity might contribute to variable-A1 formation in a cell type-specific manner.

It has been shown that diverse nuclear processes, such as gene transcription and RNA processing take place in specific compartments where many proteins and RNA molecules are concentrated [63]. For example, splicing speckles are one of the dynamic subnuclear structures containing components of the splicing machinery and other proteins, which are involved in gene transcription and 3′-end RNA processing [3,64]. Although we are unable to dissect our subcompartment A1 to splicing speckle-related loci directly, a novel technology for mapping nuclear structure does show that transcription hot zones with higher numbers of active genes, housekeeping genes, and super-enhancers are very close to RNA splicing speckles in human cells [65]. In addition, novel high-throughput image analysis for 3D structured illumination microscopy data reveals that active transcription marks, including H3K4me3 and H3K36me3, are enriched towards the RNA-filled space, but not the core of nucleosomal aggregates marked by heterochromatin signals [66]. These findings are consistent with the model proposing a transcriptional activity gradient from the nuclear periphery to the center or splicing speckles [64,65]. We observed a gradual decrease of transcription levels from subcompartment A1 to B4 (Figure 3A and Appendix A), as shown in previous studies [8,12,13], and we showed that subcompartment repositioning may confer the gene expression changes upon cell differentiation (Figure 3B and Appendix A), supporting the dynamic, reciprocal interplay between transcription and 3D genome architecture as previously proposed [67]. Indeed, genes with cell-type functions can be preferentially located in cell type-specific subcompartments in part of cell types [12], and cell type-specific genes are significantly enriched in variable A1 regions compared to housekeeping genes (Figure 5B). Moreover, we found that a subset of A1, common A1, possesses a higher gene density compared to another subcompartment of A1 (approximately two folds), and gene-rich genomic loci generally display higher transcriptional levels [68], suggesting a unique role of these genomic regions with constitutively high transcriptional activity in mammalian 3D genome organization. In support, the human chromosome with high gene density (chr19) is preferentially localized towards the nuclear interior, and the gene-poor chromosome (chr18) is predominantly positioned at the nuclear periphery [69]. In addition, studies from *Drosophila* showed active gene-rich regions are involved in defining the borders of the topological domains and forming remote intra- and inter-chromosomal interactions [70,71], suggesting a conserved role of gene-rich regions in directing the establishment of spatial patterns of chromatin interactions.

**Figure 9 biology-12-01058-f009:**
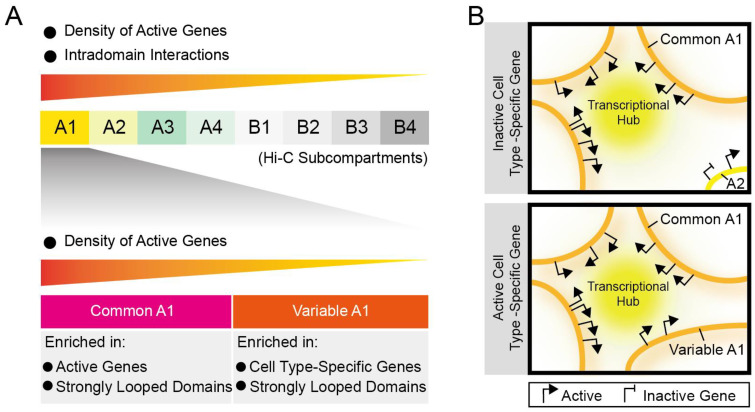
Two subsets of Subcompartment A1 are both correlated with high transcriptional activity and strong intradomain interactions. (**A**) Schematic representation of a progressive decrease of active gene density or chromatin interaction frequency within contact domains in the eight subcompartment types. On the one hand, compared to other subcompartment types, A1 regions contain the most active genes, including both housekeeping and cell type-specific genes, in ES or differentiated cells. Within A1, common A1 regions harbor approximately two-fold more active genes than variable A1, which has more cell type-specific genes. By contrast, housekeeping genes are enriched in both common and variable A1 regions. On the other hand, both common and variable A1 regions host more strongly looped domains where both housekeeping and cell type-specific genes are preferentially located, leading to more frequent interactions between regulatory elements within these chromatin domains. (**B**) A model for the transcriptional activity-driven formation of two subsets of subcompartment A1. In this model, genomic loci with very high gene density close to splicing speckles form constitutive subcompartment (labeled as common A1), and upon the increased transcriptional activity, e.g., activation of cell type-specific gene, some genomic regions move to the transcriptional hub to fall into the common A1 by transcription-dependent self-organization [1,72], e.g., the transition from A2 to A1 (labeled as variable A1) as shown.

Apart from cohesin-independent chromatin compartmentalization, the second-order mechanism of 3D genome folding is cohesin-driven loop extrusion that promotes the formation of topological domains [21,23,73,74,75]. Here, we show that both housekeeping and cell type-specific genes tend to be located within physical contact domains, and we found that strongly looped domains are most frequently present in both two subsets of A1, exhibiting a gradual decrease in domain strength from A1 to B4 (Figure 9A). Collectively, our data are consistent with a model in which genomic regions characterized by high transcription activity contribute to the formation of subcompartment A1 (Figure 9B). Genomic loci with high gene density are in close proximity to splicing speckles to form a common A1 subcompartment. Upon increased transcriptional activity, some genomic regions may move to the transcriptional hub to fall into the subcompartment A1. This special arrangement of chromatin subcompartments and intradomain interactions may, in turn, facilitate the maintenance of higher levels of gene expression, particularly for developmentally regulated genes.

## 5. Conclusions

In summary, we have identified eight subcompartments (labeled A1, A2, A3, A4, B1, B2, B3, and B4) across the genome in five distinct cell types, including ES cells, thymocytes, macrophages, NP cells, and cortical neurons. We showed that transcriptional changes of genes were closely associated with their 3D subcompartment positioning. This finding further supports that chromatin positioning in the 3D space of the nucleus is important for gene regulation during cell differentiation. A critical feature of the spatial subcompartment arrangement was that common A1, consistently identified as subcompartment A1 in all five cell types that we have investigated, had an exceptionally high gene density (1.7–2.9 folds higher than variable-A1 loci), including housekeeping genes, while cell type-specific genes were frequently located in the variable part of A1 subcompartments (odds ratios are between 1.82 and 5.36). We have demonstrated both subsets of subcompartments A1 harbored strongly looped domains where gene transcription could be regulated by cohesin-based loop extrusion. These findings presented in this study provide new insights into the mechanisms underlying spatial subcompartment A1 formation and topological domain organization in mammalian cells.

## Figures and Tables

**Figure 1 biology-12-01058-f001:**
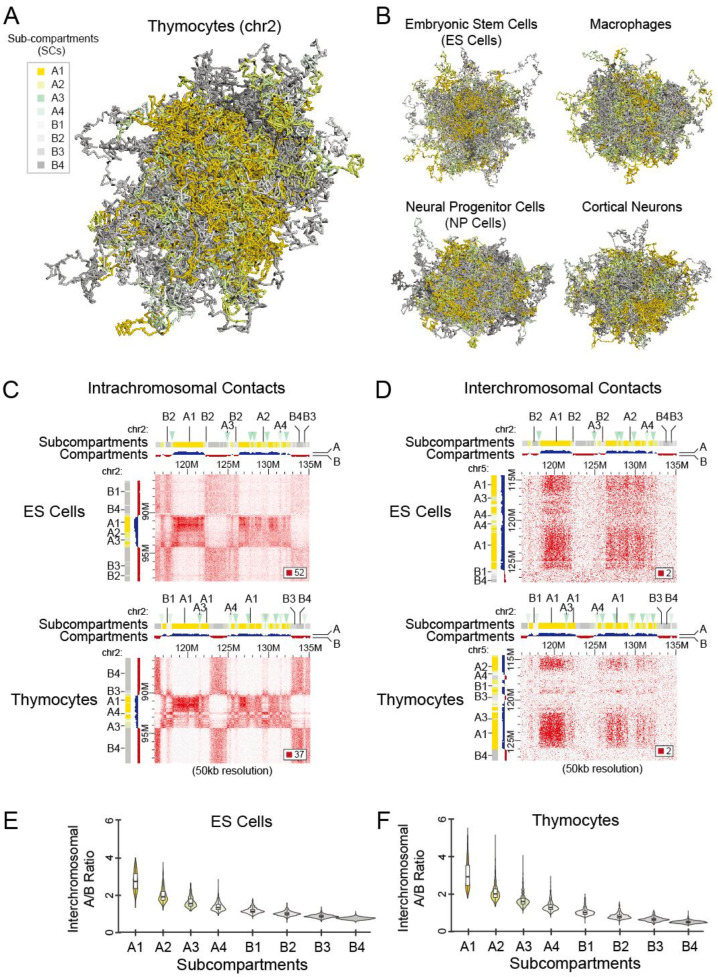
Identification of Hi-C Subcompartments. Three-dimensional plots showing chromatin interactions of the whole chromosome 2 in CD69^−^CD4^+^CD8^+^ thymocytes (**A**), as well as other four cell types containing embryonic stem (ES) cells, macrophages, neural progenitor (NP) cells, and cortical neuron (**B**). Eight subcompartment types (A1, A2, A3, A4, B1, B2, B3 and B4) were indicated in different colors. Heatmaps showing intrachromosomal (**C**) and interchromosomal (**D**) chromatin contacts in embryonic stem cells (**upper panel**) and thymocytes (**lower panel**), respectively. Violin plots show the interchromosomal A/B ratio for eight subcompartments in ES cells (**E**) or thymocytes (**F**).

**Figure 2 biology-12-01058-f002:**
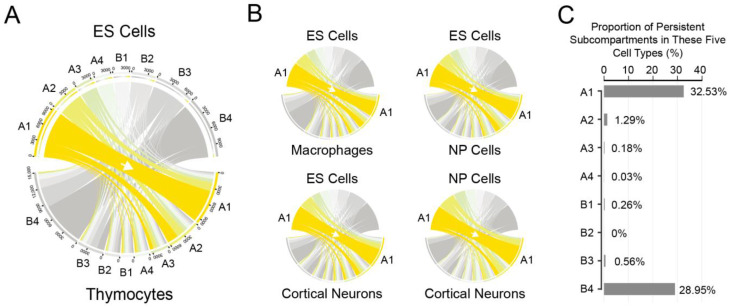
Transfers of subcompartments in different cell types. (**A**) Chord plot depicting the subcompartment transitioning from ES cells to thymocytes. (**B**) Chord plot showing repositioning from ES cells to macrophages (**upper left**), to neural progenitor (NP) cells (**upper right**), or to cortical neurons (**lower left**), as well as from NP cells to cortical neurons (**lower right**). (**C**) The proportion of subcompartment transition derived from ES cells.

**Figure 3 biology-12-01058-f003:**
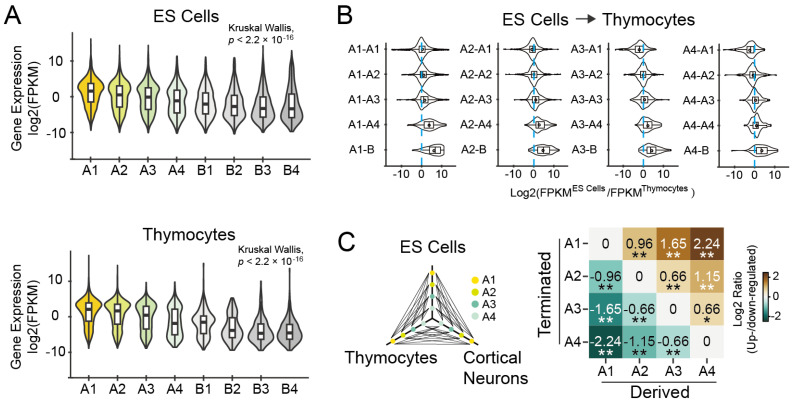
The relationship between gene expression levels and subcompartment positioning. (**A**) Violin plots showing the gene expression levels of total genes in eight different subcompartment types in embryonic stem cells (upper) or thymocytes (lower). FPKM, fragment per kilo-base exon per million mapped reads. (**B**) Relations between changes in gene expression and subcompartment switching from embryonic stem cells to thymocytes for total genes. (**C**) The diagram on the left panel shows subcompartment repositioning between cell types. Four types of subcompartments in ES cells, thymocytes, and cortical neurons, indicated by the colored points on each axis, were linked by black lines. Possibility of changes in gene expression (log 2 transformed) in response to subcompartment relocations in three distinct cell types. The horizontal axis shows derived subcompartments, and the vertical axis indicates the terminated subcompartments. * *p* < 0.05, ** *p* < 0.01 (Fisher’s Exact Test).

**Figure 4 biology-12-01058-f004:**
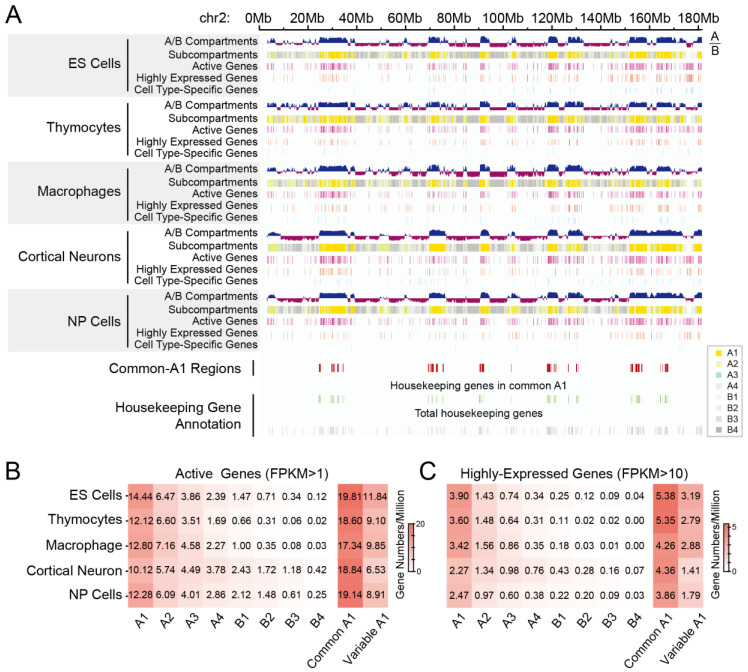
Gene density in different subcompartment types. (**A**) Chromatin compartment annotations of the whole chromosome 2, including A/B compartments and subcompartments. Transcription start sites of active genes (FPKM > 1, purple), highly expressed genes (FPKM > 10, red), cell type-specific genes (cyan), housekeeping genes in common-A1 regions (green), and total housekeeping genes (gray) were highlighted. Gene numbers per million of active genes (**B**) or highly expressed genes (**C**) in each subcompartment type across diverse cell types. Common A1 is consistently identified as subcompartments A1 in all five cell types. Variable A1, subcompartments A1 but apart from common A1.

**Figure 5 biology-12-01058-f005:**
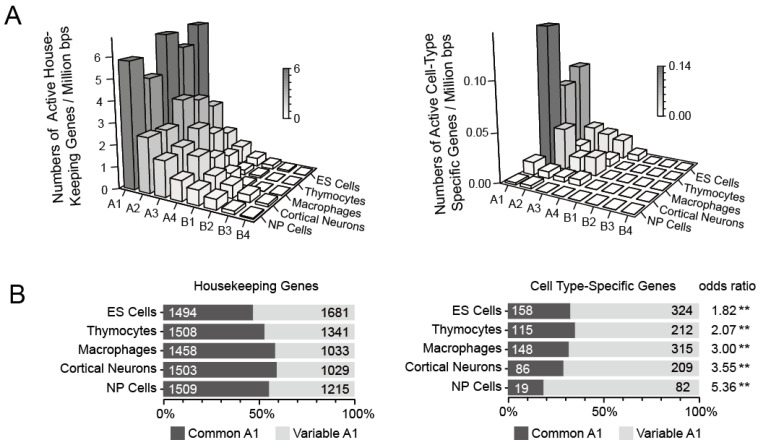
Gene density of different subcompartments in five cell types. (**A**) Bar plots showing gene density of housekeeping genes (**left panel**) or cell-type specific genes (**right panel**) in eight subcompartments in these five cells. (**B**)The relative distribution of housekeeping genes (**left panel**) or cell type-specific genes (**right panel**) in common A1 and variable A1 regions. Gene numbers were indicated on the bar plots. ** *p* < 10^−8^ (Fisher’s Exact Test).

**Figure 6 biology-12-01058-f006:**
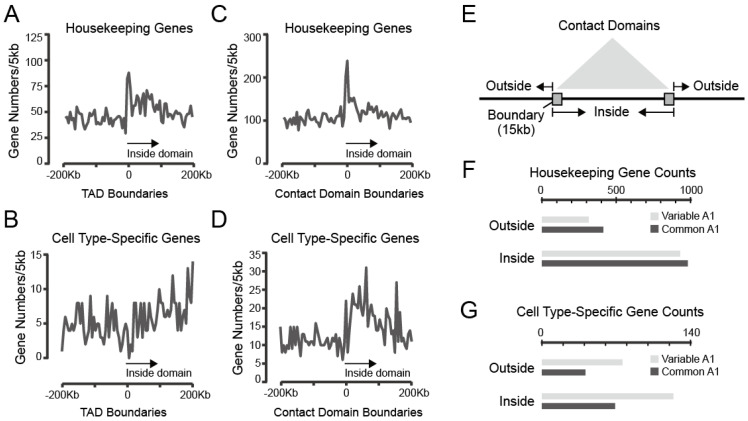
Both housekeeping and cell type-specific genes are enriched in contact domains in thymocytes. (**A**,**B**) TADs boundaries were associated with housekeeping but not cell type-specific genes. (**C**,**D**) Contact domains were enriched for both housekeeping and cell type-specific genes compared to adjacent regions. (**E**) The diagram shows the inside and outside regions of contact domains. Bar plots showing gene numbers of housekeeping (**F**) or cell type-specific genes (**G**) within contact domains or outside domains in common A1 or variable A1 subcompartments, respectively.

**Figure 7 biology-12-01058-f007:**
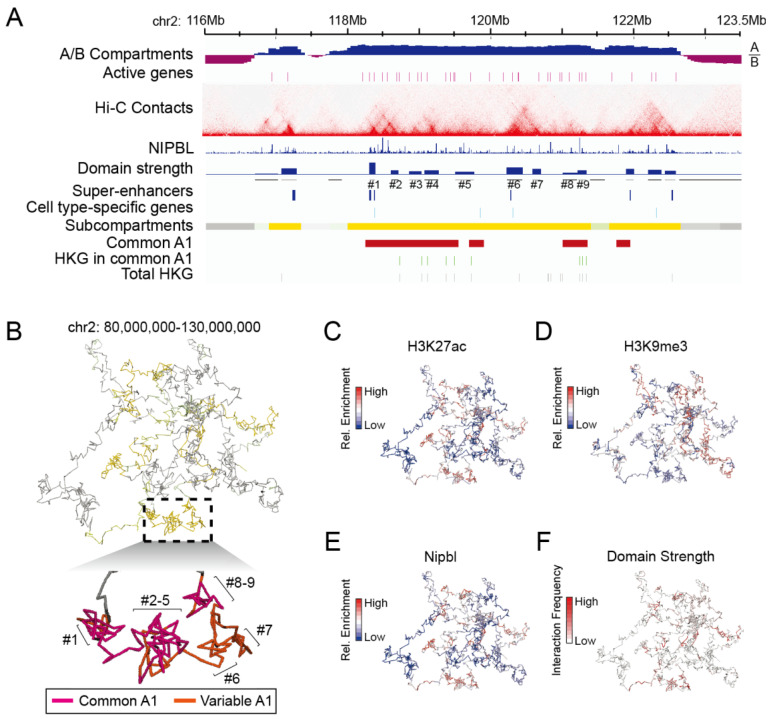
Strongly organized contact domains are frequently located in subcompartments A1, enriched in H3K27ac and Nipbl signals. (**A**) Hi-C contact maps in a region on chromosome 2 (chr2: 116,000,000–123,500,000) showing contact domains in thymocytes at 5 kb resolution aligned to the annotated tracks of A/B compartments, Nipbl occupancy, domain strength, super-enhancers, subcompartments, and common A1 (indicated by red). Transcription start sites of active genes (purple), cell type-specific genes (cyan), housekeeping genes in common-A1 regions (green), and total housekeeping genes (gray) was indicated by different colored lines. Nine contact domains were indicated by #1–#9, shown in the panel (**B**). (**B**) 3D simulation of chromatin interactions illustrates the relationship between subcompartment positioning and contact domain strength. The inset (lower panel) showed an enlarged plot of contact domains 1–9. Chromatin associated with common A1 was indicated in purple, and variable A1-related chromatin was indicated in red. Chromatin is marked by signals of H3K27ac (**C**), H3K9me3 (**D**), Nipbl (**E**), or contact domain strength (**F**), respectively.

**Figure 8 biology-12-01058-f008:**
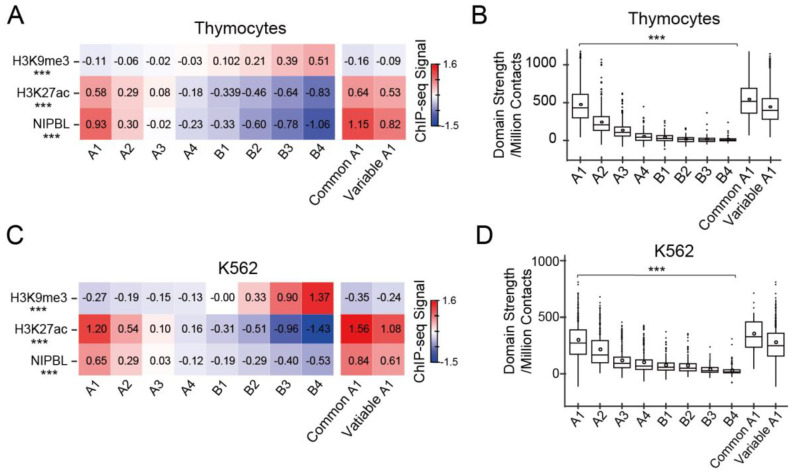
Contact domain strength in different subcompartment types in both mouse and human cells. Enrichment analysis of histone modification H3K9me3 and H3K27ac, as well as Nipbl signals in each subcompartment type across the genome in thymocytes (**A**) and human K562 cells (**C**). Contact domain strength in eight subcompartments, as well as in common and variable-A1 regions in thymocytes (**B**) and human K562 cells (**D**). The outliers were indicated by filled dots, and means denoted by open circles on the boxplot. *** *p* < 10^−16^ (Kruskal-Wallis Test).

## Data Availability

High throughput sequencing data for this paper are available from the NCBI Gene Expression Omnibus (GEO) under accession number GSE227868.

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
