# Peer review of "Identification of Two Subsets of Subcompartment A1 Associated with High Transcriptional Activity and Frequent Loop Extrusion"

_biology, 2023, doi:10.3390/biology12081058_

Round 1

Reviewer 1 Report

The manuscript by Yin and co-authors "Identification of Two Subsets of Subcompartment A1 Associated with High Transcriptional Activity and Frequent Loop Extrusion" is focused on investigation of the relationship between chromatin compartment organization and gene activity and underlying mechanisms of compartment formation. The authors identified two subcompartment types belonging to the A1 subcompartment, which vary in gene content and tissue-specific expression pattern and showed that gene activation is correlated with 3D subcompartment rearrangement. These findings are inline with previous results linking gene activity and spatial organization of chromatin. The manuscript can be accepted for publication after the authors properly address my critical points. 

Major points:

1) In modeling 3D chromosome structure based on chromatin contact maps, interactions of chromosomal DNA with other nuclear anchors like nuclear lamina and interchromosomal contacts were apparently disregarded. How one can be sure these external interactions do not affect chromatin folding and subcompartment formation?

2) The authors claim that gene expression is influenced by subcompartment positioning, but since they do not study the coordinated time-course of the subcompartment rearrangement and gene expression, the causal connections between the two processes is rather hard to justify. Maybe they have to merely state the correlation?

3) Fig.7. I'm a bit confused about the modeling of chr.2. In zoomed-out section of Fig.7B, domain #1 is visually larger than domain #8-9, while on the Fig7,A they show at least 2-fold difference in bp. Please, explain.

Minor points:

1) line 67:  While initially six subcompartments were identified, later on the authors mention only five of them (line 71: "In contrast, the other three subcompartments (B1, B2, and B3) are correlated with genomic loci in compartment B, containing the tightly packed form of DNA"). Please, be consistent.

2) I do not quite understand why the authors in their CUT&Tag experments use secondary gouat-anti Rabbit antibodies. The affinity of pG-Tn5 transposase is much higher for rabbit IgGs and can be partially blocked by the secondary low-pG affinity goat IgG binding. Please, explain.

3) In Fig.1(A,B) (and throughout the manuscript) the color scheme for subcompartment indication should be changed for better visual discrimination of subcompartments. Personally I do not see the difference between, say, A4, B2 and B3. A more contrast LUT should be applied.

4) Fig.1(E,F). Could you apply the same Y-axis scaling for both plots?

5) Fig.8(A,B). Calibration bars for these plots should have the same scale for easier comparison.

6) Fig.8(D). What is the y-axis type? Definitely neither linear nor log.

There are some typos or weird phrases that should be corrected. For example:

1) line 37:  "determinants" - "determinant"?

2) line 62: "During cell differentiation, chromatin undergoes from compaction to decompaction process" - "During cell differentiation, chromatin undergoes decompaction"? Moreover, during cell differentiation chromatin rather undergoes global compaction with some tissue-specific genes being activated and decompacted.

3) line 103: "in five different cells" - "in five different cell types"?

4) line 282: "3.1.2 Frequent occurrences of subcompartment relocations in these five cells" - I'd rather suggest the section title to be rephrased, like "Frequent occurencies of subcompartment relocations during cell differentiation"

Reviewer 2 Report

The manuscript by Yin et al., studied the relationship between gene
transcriptional activity and their spatial subcompartment positioning. The authors used a genome-wide Hi-C analysis and identified  eight types of subcompartments (A1, A2, A3, A4, B1, B2, B3, and B4) in different cell types. This topic of the study is interesting and the study tries to answer a fundamental question, however, the study suffers from a few issues. Here are my points to the authors:
1- How did the authors define these eight subcompartments? Why 8 not 7 or 6 as identified from other studies? Is there a statistical analysis that can see the difference between these identified 8 domains? 2- What is teh effect of the cell cycle of the  subcompartmentsidentifcation? The authors should provide a control experiment showing the effect of the cell cycle on the HiC analysis.
3- How can the authors relate these 8 subcompartments to other classifications based on fluorescence imaging? For example, a recent study by Miron et al., 2020, Science Advances showed a robust classification for the chromatin compaction degrees, up to 3 classes. That also was related to gene expression, It would be very important that the authors discuss their findings in relation to this study?

It is good written, English can be slightly improved

Reviewer 3 Report

This article represents an interesting exploration of chromatin subcompartments using publically available Hi-C, RNA-seq and ChIP-seq data from murine embryonic stem cells and 4 terminally differentiated murine cell types. The authors then demonstrate the broader relevance of their findings by comparison with two human immortalised cell lines, generating some of their own epigenetic data in order to do so.  

The introduction lays a good foundation for this work, by describing our current understanding of the relationship between transcriptional activity and chromatin compartmentalisation. The analyses performed make use of appropriate published tools, for example, inferring subcompartments (Calder) and generating 3-dimensional models of chromatin (WashU Epigenome browser) using the aggregate contact frequencies present in Hi-C data.

Gene density and expression levels are shown to differ between subcompartments and changes in the inferred subcompartment between cell types are observed and used to discriminate between genomic regions that are consistently or variably associated with the gene-dense subcompartment (A1).

In summary, the authors show how their observations support prevalent concepts regarding the mechanisms that govern chromatin architecture, including the role of transcription in mediating compartmentalisation. The paper represents a thorough exploration of the subcompartments called by Calder, similar to that already included in the original Calder publication.

Whilst thorough and well described, the novelty of the observations made is occasionally overstated, and their predictability is downplayed. For example Calder uses a k-means approach to infer k=8 subcompartments and, as I understand, A.1.1 (referred to as A1 in this manuscript) is by definition the subcompartment with the highest gene density. Therefore, it is self-fulfilling that 8 compartments were identified and that they differ based on gene density. It is also unsurprising that they differ based on transcriptional activity, given that gene dense (A) compartments are known to have higher transcription than gene poor (B) compartments.

A main observation of the study is that consistently inferred compartments that contain the highest gene density (common A1) contain constitutively transcribed genes (housekeeping genes). As the authors mention, these compartments are partly established and maintained by transcriptional processes, it is therefore logical that they should not change in regions with constitutive gene expression. Gene density does not change between cell types and housekeeping genes are often found in gene dense regions.

If the authors were to demonstrated convincingly that the subcompartments discussed in this paper represent truly distinct chromatin states, which are subject to different biological processes it would dramatically improve the impact of the manuscript. At present it seems that each subcompartment represents a division of a gradual spectrum, as illustrated in figure 9..

Specific recommendations:

1.       In general the manuscript is very comprehensible, however improvements can be made to the readability of the abstract. A suggested rewrite is included below. In general changes are purely linguistic, with the exception of the second sentence, which states that “we… identify eight types of subcompartment”, which reads as though the number eight is not predetermined and represents a result of this work. Similarly I have sought to clarify that A1 compartments are defined by high-gene density by the Calder algorithm (see also subsequent recommendation):

Abstract: 3D genome organization has been increasingly recognized as an important determinants of the precise regulation of gene expression in mammalian cells, yet the relationship between gene transcriptional activity and their spatial subcompartment positioning are still not fully comprehended. Here, we first utilized genome-wide Hi-C data to identify infer eight types of subcompartments (A1, A2, A3, A4, B1, B2, B3, and B4) in mouse embryonic stem cells and four primary differentiated cell types, including thymocytes, macrophages, neural progenitor cells, and cortical neurons, respectively. To a large extent, tTransitions of subcompartments may confer the gene expression changes in different cell types. Intriguingly, we identified two subsets of subcompartments characterized defined by higher gene density and characterized by strongly looped contact domains, named common A1 and variable A1, respectively. We revealed that the common A1, including which includes highly expressed genes and abundant housekeeping genes, shows a ~2-fold higher gene-density than the variable A1 where cell type-specific genes are significantly enriched in. Thus, our study supports a model in which both types of genomic loci with constitutive or regulatory high transcriptional activity can drive the subcompartment A1 formation. Special chromatin subcompartment arrangement and intradomain interactions may in turn contribute to maintaining proper levels of gene expression, especially for regulatory nonhousekeeping genes.

2.       The Calder algorithm represents a crucial component of the methodology implemented in this paper, therefore, I feel it is important to briefly mention that the number of subcompartments is predetermined, that their identity is defined based on gene-density, and that certain epigenetic marks, and transcriptional activity, have already been demonstrated to correlate with inferred subcompartments. An appropriate location to include this could be in the introduction, immediately after the identification of six different subcompartments is mentioned on line 67, pg. 2. I would also invite the authors to clarify the novelty of their findings compared to those made in the original Calder publication

3.       In section 3.1.2. it is not clear exactly how repositioning is defined and what the biological significance of this might be .

a.       This should be clearly defined: For example, if the boundaries of a subcompartment are altered only slightly between cell types does this count as repositioning, as this is unlikely to be biologically significant. An alternative approach that may be more informative would be to segregate the genome into bins of equal size, assign a subcompartment identity (e.g. by greatest overlap) and report the proportion of bins that change compartment.

b.       The authors should clarify what biological significance they ascribe to subcompartment repositioning. When reported as a whole this includes both small (A1-A2) and large (A1-B4) transitions, however the significance of these transitions is likely to differ greatly. Large transitions appear relatively rare and so reporting the total number of transitions (or total number of subcompartments that do not change as per Fig 2B) conveys little meaning and exaggerates the changes observed. It would be useful to report statistics for larger transitions, e.g. A-B.

4.       In the text (e.g. line 345, pg. 9) the authors restrict some observations to chromosome 2, presumably because this is what is shown in Fig. 4A, etc.. The authors should clarify whether this is a representative example of the whole genome and whether other analyses, including that mentioned on line 347, pg 9 and shown in Fig 4B incorporate the whole genome.

5.       On line 374, pg 10 the authors state that “housekeeping genes were enriched in subcompartments [sic] A1 significantly”, but it is not clear how this has been established statistically.

6.       On line 455, pg. 13 a suggested correlation between Nipbl occupancy and subcompartment positioning is mentioned. This should be tested statistically, as should potential correlation with domain-strength , H3K27ac and H3K9me3 occupancy.

7.       I was unable to understand what is shown in Figure 3, panel C. Could the authors clarify what is shown by the triangular plot with ES Cells, Thymocytes and Cortical Neurons shown on each axis, as well as what is meant by the “Terminated” and “Derived” labels on the stochastic matrix

8.       Figure 9B includes a model for transcriptional activity-driven formation of subcompartments, which includes reference to speckles. Given that nuclear speckles do not form any part of this manuscript this feels like a bit of a reach. Similarly the reference to splicing speckles in the text is very specific. A reference to transcriptional machinery/hubs more generally would feel more appropriate.

9.       In the conclusion (line 583, pg. 17) the high-gene density of the common A1 subcompartment is described as “surprising”. This is not the case, in fact the Calder algorithm identifies these compartments as A1 based on gene-density.

Minor comment

1.       In the Simple summary (line 30) common A1 is referred to as “constitutive subcompartment A1”, in contrast to the remainder of the manuscript. Consistent terminology should be used.

2.       In figure 7A the Hi-C contacts and Sub-compartments tracks do not seem to correspond very well (domains visible in the Hi-C data do not seem to line up with sub-compartments), which does not engender confidence in a crucial component of this study.  This should be explained.

Very minor improvements to the quality of English language can be made throughout the text, where these are important they have been included in the comments and suggestions for authors.
